# Aggressive behaviour of psychiatric patients with mild and borderline intellectual disabilities in general mental health care

Jeanet Grietje Nieuwenhuis[1]*, Peter Lepping[2,3,4], Cornelis Lambert Mulder[5], Henk Liewellyn Inge Nijman[6,7†], Eric Onno Noorthoorn[8]

1 Department VGGNet, Ggnet Mental Hospital, Warnsveld, Netherlands, 2 Betsi Cadwaladr University Health Board, Wrexham, Wales, United Kingdom, 3 Mysore Medical College & Research Institute, Mysuru, India, 4 Department Centre for Mental Health and Society, Wrexham Academic Unit, Wrexham, United Kingdom, 5 Department Psychiatry, Cornelis Lambert Mulder Erasmus University Rotterdam, Rotterdam, Netherlands, 6 Department Behavioural Science Institute, Radboud University Nijmegen, Nijmegen, Netherlands, 7 Firvoor Forensic Hospital, Den Dolder, The Netherlands, 8 Department Research and Development Vordenseweg Ggnet Mental Hospital, Warnsveld, Netherlands

† Deceased.
* j.nieuwenhuis@ggnet.nl

**Data Availability Statement:** All relevant data are available in the paper, its Supporting Information files, and on OSF: https://osf.io/7vejy/.

## Abstract

### Purpose

Little is known about the associations between mild intellectual disability (MID), borderline intellectual functioning (BIF) and aggressive behaviour in general mental health care. The study aims to establish the association between aggressive behaviour and MID/BIF, analysing patient characteristics and diagnoses.

### Method

1174 out of 1565 consecutive in-and outpatients were screened for MID/BIF with the Screener for Intelligence and Learning Disabilities (SCIL) in general mental health care in The Netherlands. During treatment, aggressive behaviour was assessed with the Staff Observation Aggression Scale-Revised (SOAS-R). We calculated odds ratios and performed a logistic and poisson regression to calculate the associations of MID/ BIF, patient characteristics and diagnoses with the probability of aggression.

### Results

Forty-one percent of participating patients were screened positive for MID/BIF. Patients with assumed MID/BIF showed significantly more aggression at the patient and sample level (odds ratio (OR) of 2.50 for aggression and 2.52 for engaging in outwardly directed *physical* aggression). The proportion of patients engaging in 2–5 repeated aggression incidents was higher in assumed MID (OR = 3.01, 95% CI 1.82–4.95) and MID/BIF (OR = 4.20, 95% CI 2.45–7.22). Logistic regression showed that patients who screened positive for BIF (OR 2,0 95% CL 1.26–3.17), MID (OR 2.89, 95% CI 1.87–4.46), had a bipolar disorder (OR 3.07, 95% CI 1.79–5.28), schizophrenia (OR 2.75, 95% CI 1.80–4.19), and younger age (OR

**Funding:** The author(s) received no specific funding for this work.

**Competing interests:** One of the authors of this article (H. Nijman) is a (co-)author of both the aggression scale (SOAS-R) and the Screener for Intelligence and learning disabilities (SCIL) that were used to collect data in this study. The authors have declared that no competing interests exist. This does not alter our adherence to PLOS ONE policies on sharing data and materials.

1.69, 95% CI 1.15–2.50), were more likely to have engaged in any aggression. Poisson regression underlined these findings, showing a SCIL of 15 and below ($\beta = 0.61$, $p<0.001$) was related to more incidents.

## Conclusions

We found an increased risk for aggression and *physical* aggression in patients with assumed MID/BIF. We recommend screening for intellectual functioning at the start of treatment and using measures to prevent and manage aggressive behaviour that fits patients with MID/BIF.

## Introduction

Mild intellectual disability (MID) and borderline intellectual functioning (BIF) are highly prevalent in general mental health care but often stay unnoticed [1, 2]. Our research group has previously shown that in the Netherlands, the prevalence of MID/BIF increases by setting, from 27% in outpatient settings, to 40% in Flexible Assertive Community Treatment (FACT) teams and admission wards, to 67% in long-stay wards [3]. Furthermore, in the admissions wards, patients with MID or BIF were found to have increased risks of having been involuntarily admitted in the past (OR 2.71) and being subjected to coercive measures (OR 3.95) [1]. Aggressive and dangerous behaviours are the main reason for involuntary admissions and seclusion in the Netherlands. The severity and dangerousness of disruptive behaviour perceived by treating staff influence the decisions to use restrictive measures [4]. These measures are widely recognised as interventions that potentially have severe negative consequences for the patient, including trauma [5]. Aggression is often called "challenging behaviour" (CB) in intellectual disability (ID) services, and the use of coercive measures also has a significant impact on staff and healthcare workers. On average, 62% of nurses in different countries indicate they have experienced physical violence over the course of a year [6]. Health care workers [7–9] and workers in ID services [10, 11] experience psychological and emotional consequences of aggression such as post-traumatic stress, depression, and a negative impact on work functioning and job satisfaction.

From studies in institutions for people with ID [12, 13], we know that CB is a common problem. However, Bowring and colleagues [13] noted no agreed consensual, conceptual, or operational definition of CB. In population studies, considerable variation in CB prevalence is found (4%-22%, [14]). Communication problems, the severity of the ID, and psychopathology are associated with a higher risk of CB [14–16]. In a large Dutch study of an inpatient ID service covering 421 patients, 20% of the patients involved in aggression incidents were responsible for 50% of the verbal and 80% of the physical, aggressive incidents [17]. This study showed that the more severe the disability, the higher the possibility of repeated incidents in a single patient. Such patterns of incidence showing repetitive aggression in patients with more severe intellectual disability can also be expected to occur in general psychiatry.

A review of 424 studies conducted in general psychiatry in various settings across 11 countries showed that 32.4% of patients admitted to psychiatric facilities engaged in aggressive behaviour or violence and generated 182.8 events per 100 admissions [6]. Studies also show that a small subgroup of patients is generally responsible for a large proportion of violent incidents [6, 18]. Many previous publications have been single-centre reports, making comparison and generalising conclusions difficult. One indirect measure of aggression that considers

whole country data is the UK NHS staff survey. In 2019, 48% of the 1.1 million NHS staff participated in the survey. 14.7% of all respondents reported having personally experienced violence from patients, relatives, or public members. This figure rose to 20.2% in the staff working in mental health services and 34% in those working for the ambulance services. It decreased to as low as 5.5% in acute services and 7.5% in non-psychiatric community services. Five years trends are remarkably stable in all measured groups [19]. Aggression and coercive measure are closely linked. Using whole country data, coercion figures were remarkably similar across four European countries [18]. An analysis of Welsh coercion data from this study across all Welsh Health Boards demonstrated twice as many coercive measures when ID services are included (2013 total incidents, Wales: 3735) compared to when ID services are not included (2013 incidents without ID, Wales: 1886). The results also showed that the number of patients affected by coercive measures per 100 occupied bed days was not affected by adding the ID data, but the number of coercive measures was. This suggests that those patients with ID who were affected by coercive measures were coerced multiple times and more often than the non-ID population [20]. This is similar to what we know from aggression data.

In a study on admission wards [1], we showed that patients with BIF/ MID had an increased risk of involuntary admission (OR 2.71; SD 1.28–5.70) and coercive measures (OR 3.95, SD 1.47–10.54). These findings were confirmed in nationwide data gathered in 2014, where intellectual impairment also showed an association with increased risk of seclusion and other coercive measures [21]. Internationally, there is evidence that patients with BIF/ID account for more and more prolonged seclusion and restraint events [2, 3].

Until now, however, the level of cognitive function has hardly been studied as a potential 'predictor' of aggression, although MID /BIF is much more prevalent in general mental health care than previously assumed [3]. Therefore, in this study, we examined the associations between MID/BIF and aggressive behaviours in a sample of psychiatric inpatients and outpatients. We hypothesised that:

1. In mental health care services, patients suspected to have MID/BIF are more often engaged in aggression incidents

2. Patients suspected to have MID/BIF are more often involved in outwardly directed *physical* aggression and have more incidents per person than patients not suspected to have MID/ BIF patients.

## Method

### Setting

We collected a consecutive sample of patients treated with four different types of care in a mental health care trust in the east of the Netherlands, covering a catchment area of 630000 inhabitants.

These four types of care concerned were:

1. Outpatient psychiatric clinics, in this context, are the services the general practitioner refers to patients for initial mental health care. This service provides acute crisis interventions, outpatient psychological and psychiatric treatment, and support.

2. Flexible Assertive Community Treatment (FACT) teams specialised in daily (outpatient) support and treatment for patients with serious mental illness (SMI). In the Netherlands, FACT teams are multidisciplinary outpatient teams with 8–10 professionals, such as psychiatrists, psychologists, nurses, and social workers, generally caring for 200 patients with SMI.

3. General admission wards admit first-onset patients and patients referred from FACT teams or outpatient clinics. Patients at these wards were eligible for inclusion in the current study when they resided on the ward for at least six days.

4. Long stay wards, providing residential care for patients with SMI. Patients all have a long history of receiving professional support and treatment, primarily in FACT teams.

The study was conducted and reported in accordance with the STROBE guidelines for reporting observational studies [22]. Screening for potential ID and data collection for aggressive incidents was done from May 2014 until January 2019. All patients treated in participating settings were asked to join the study to screen for potential IDnd participants who agreed to participate provided written informed consent for this.

## Measures

The Staff Observation Aggression Scale-Revised (SOAS-R) was used to register aggression and is a widely used instrument to document the nature and severity of aggressive incidents. The SOAS-R records the following five aspects of aggressive incidents: (a) the apparent provocation, which led to the aggressive event, (b) the means used by the patient during the aggressive event, (c) the target of aggression, (d) the consequence(s) for the victim(s) of the aggression, and (e) the measures taken to stop the aggression, such as seclusion.

The inter-observer reliability of SOAS and SOAS-R aggression observations is acceptable, with a Cohen's kappa of 0.61 and 0.74, respectively, and a Pearson product-moment correlation coefficient between independent raters of 0.87 [23]. The SOAS and SOAS-R severity scores correlate significantly with various other aggression measurement methods (i.e., correlations from 0.38 to 0.81) [24]. The scale is quick to complete, and there is no need for staff to be trained to use it.

We used the SCreener for Intelligence and Learning disability (SCIL) to detect patients with MID or BIF [23, 24]. Translation for use in English is in preparation. The SCIL is a test consisting of 14 questions, including educational level and small tasks intended to screen for patients' overall cognitive abilities [24]. It was developed specifically to detect MID/BIF (IQ 50–85) in people in a range of settings, such as (mental) healthcare or social service settings and police stations and shelters for people experiencing homelessness. The reliability of the SCIL, as expressed in Cronbach's alpha in the initial validation study, was good (0.83 in 318 adult subjects). The AUC value for detecting MID/BIF was 0.93, which is excellent. With 19 or lower as a cut-off score, the SCIL accurately classified 82% of people with MID/BIF. Of the ten people without MID/BIF, 9 (89%) were classified correctly as having no MID/BIF. In accordance with the SCIL manual, administering the SCIL requires no specific clinical skills.

The SCIL has recently been validated in patients with SMI in FACT teams [25]. The Cronbach's alpha of the SCIL in that sample was 0.73. The AUC value for detecting MID/BIF and MID was 0.81, with percentages of correctly classified subjects of 73% and 79%, respectively. We used two cut-off scores: 19 and 15. Above 19 implies no MID/BIF, and 19 and below implies a (suspected) MID/BIF. The cut-off point of 15 and below implies a (suspected) MID [26]. In the following descriptions, we use two cut-off points, 19 for MID or BIF and 15 for MID only. The SCIL assessments used in the current study were performed between 2014 and 2018 [3]. We included all SOAS-R incidents reported in routine care between 2014 and 2019.

Patients were excluded from screening for potential ID with the SCIL based on (1) an inadequate grasp of the Dutch language, (2) lack of cooperation, (3) an inability, in the assessor's opinion, to concentrate for at least 20 minutes in order to engage in the test as outlined in the instruction [26].

Nurses in inpatient and outpatient settings were trained to administer the SCIL. According to the questionnaire instructions, the SCIL was administered by a person not involved in the treatment. In the mental health trust where the study was carried out, the SOAS-R has been used since 2007 as a standard tool for nurses to log incidents and medical incident reports in inpatient and outpatient settings.

Demographic data and diagnosis were extracted from the electronic medical charts (EMC): age, gender, psychiatric diagnosis (DSM-IV-TR, as assessed by the psychiatrist), and Global Assessment of Functioning (GAF) score.

## Statistical analyses

At the level of the patient, we identified whether a patient had shown an aggressive incident and whether a patient had shown outwardly directed *physical* aggression incidents against persons (so not against themselves). The total number of SOAS-R incidents per patient reported between 2014 and 2019 was also counted. Differences in the number of incidents between patients with or without MID/ BIF were tested using the Kruskal-Wallis rank order test because of extremely skewed frequencies. As mentioned earlier, the SCIL outcomes were categorised in scores of 19 and less, representing assumed MID/BIF and scores of 15 and less, representing assumed MID. BIF, MID and patient characteristics were cross-tabulated with having shown aggression incidents and *physical* aggression incidents against persons. We calculated chi-square statistics and Odds ratios to investigate the significance of the differences and the increased risk of showing (*physical*) aggression in relation to patient characteristics.

We also performed a logistic regression analysis to understand the association of these variables with having shown any aggression or *physical aggression* corrected for one another. A forward entry and backward deselection procedure were used. All variables selected from the EMC were entered in the analysis. Thus gender, age categories, diagnosis, MID or BIF as assessed with the SCIL. For the forward selection, variables with associations having a p-value of <0.2 were included in the logistic regression analysis, following the relevance criterion proposed by Hosmer and Lemeshow [27]. These were entered in 3 blocks: the demographic variables, the diagnoses, and the response categories in the SCIL.

Next, Poisson regression was applied to the number of incidents as we may expect a skewed distribution, and the number of incidents represents a count. Before applying the regression, the distribution of the number of incidents was tested. We applied forward entry and backward deselection to investigate which patient characteristics predicted the number of aggression incidents. We present the β, which as a rate ratio can be interpreted as a growth or downturn rate [28].

## Ethical considerations

Ethical approval for the study was provided in 2014 by the ethical board of the University of Twente, Enschede, The Netherlands. All procedures performed in the current study were in accordance with the Helsinki Declaration of 1975, revised in 2008, and with comparable ethical standards. Data were analysed based on fully anonymised data that allowed none of the cases to be traced to an individual.

## Results

### SOAS-R score in general

In total, we found 1472 aggressive incidents in 196 (16.7%) of the 1565 patients. Most of the registered incidents occurred in inpatients. Only 36 outpatients were involved (18.3% of the

196, 2.2% of the complete sample). Of the 196 patients with an incident of aggression, 47 were involved in one incident, 84 patients between two and five incidents, and 65 were involved in over six incidents. 23 (11.7% of 196) patients were responsible for 751 aggression incidents (51.0% of 1472). The mean number of incidents was 7.53 per patient, with a maximum of 78 incidents. Of the 1565 patients, 105 patients were engaged in 269 physical, outwardly aggressive incidents (18.3% of the 1472 incidents). Of these 105 patients, 46 were involved in one incident, 51 in between two and five, and 8 in over six physically aggressive incidents. 20 (7.4%) of these patients were responsible for 137 (50.9%) of the 269 incidents. Both analyses show that approximately 10% of the patients account for half of the aggression incidents.

## Sample and SCIL

We asked 1565 consecutive patients to participate. We obtained a SCIL score in 1174 cases (75.0%). 481 (41.0%) of the 1174 included patients showed a SCIL score of 19 and below (assumed MID/BIF). 239 (20.4%) showed a SCIL score of 15 and lower (assumed MID). In the various settings, the response was comparable with 71.5% at the outpatient services, 73.1% at the FACT teams, 75.5% at the long-stay wards and 78.9% at the admission wards [3]. The distribution of diagnoses was comparable in the participants compared to the non-responders, discarding selection bias by diagnosis.

## SOAS-R and SCIL score, univariate analyses

Table 1 presents the number of aggression incidents over the SCIL negative or positive groups for MID and BIF. It shows that the proportion of patients engaging in (repeated) violent behaviour, in general, is higher in patients assumed to have MID or BIF. Furthermore, the table indicates that outwardly directed *physical* aggression occurred more often in patients with assumed MID. The odds ratios show that these increase in the higher categories above two incidents per patient. In general, the odds ratios for MID are higher than those for BIF, implying that an increasing number of incidents is associated with BIF but even more frequent in MID patients (Table 1.)

Table 2 presents the SCIL outcomes, patient characteristics, and aggression frequencies. A SCIL outcome of 19 and below (assumed BIF or MID) was associated with more aggression in general (OR = 2.50), as well as with more *physical* aggression (OR = 2.52). A SCIL outcome of

**Table 1. BIF and MID compared to the number of aggression incidents.**

| | | N = | Aggression in general | | | | | *Physical* aggression | | | | |
| --- | --- | --- | --- | --- | --- | --- | --- | --- | --- | --- | --- | --- |
| | | | No aggression | One incident | 2–5 incidents | > 5 incidents | P = | No aggression | One incident | 2–5 incidents | > 5 incidents | P = |
| SCIL | above 19 | 693 | 636 (91.8%) | 18 (2.6%) | 19 (2.7%) | 20 (2.9%) | <0.001 | 665 (96.0%) | 13 (1.9%) | 12 (1.7%) | 3 (0.4%) | <0.001 |
| | 19 and below (BIF/MID) | 481 | 385 (80.0%) | 20 (4.2%) | 51 (10.6%) | 25 (5.2%) | | 432 (89.8%) | 24 (5.0%) | 23 (4.8%) | 2 (0.4%) | |
| OR | category / else | | 0.24 (0.17–0.36) | 1.62 (0.85–3.11) | 4.20 (2.45–7.22) | 1.84 (1.10–3.35) | | 0.37 (0.22–0.59) | 2.74 (1.38–5.45) | 2.84 (1.40–5.78) | 0.96 (0.16–5.77) | |
| P = | | | < 0.001 | 0.148 | < 0.001 | 0.045 | | < 0.001 | 0.004 | 0.004 | 0.965 | |
| SCIL | above 15 | 935 | 839 (89.7%) | 25 (2.7%) | 41 (4.4%) | 30 (3.2%) | <0.001 | 903 (96.6%) | 8 (0.9%) | 9 (1.0%) | 3 (0.3%) | <0.001 |
| | 15 and below (MID) | 239 | 182 (76.2%) | 13 (5.4%) | 29 (12.1%) | 15 (6.3%) | | 222 (92.9%) | 16 (6.7%) | 14 (5.9%) | 2 (0.8%) | |
| OR | category / else | | 0.36 (0.24–0.51) | 2.09 (1.05–4.15) | 3.01 (1.82–4.95) | 2.02 (1.06–3.81) | | 0.46 (0.25–0.84) | 8.31 (3.51–19.67) | 6.40 (2.73–14.97) | 2.62 (0.43–15.77) | |
| P = | | | < 0.001 | 0.037 | < 0.001 | 0.030 | | 0.013 | < 0.001 | < 0.001 | 0.292 | |

**Table 2. Association between aggression BIF, MID and patient characteristics.**

| Predictors | | Aggression incidents | | | OR | 95% CI OR | Physical aggression | | | OR | 95% CI OR |
|---|---|---|---|---|---|---|---|---|---|---|---|
| | | No aggression | Aggression | P = | | | No aggression | Aggression | P = | | |
| N$_{patients}$ = | | 1369 | 196 | | | | 1460 | 105 | | | |
| Age | | 43.3 | 42.3 | 0.290 | | | 43.2 | 41.9 | | | |
| Age categories[1a1] | 0–35 | 407 (29.7%) | 73 (37.2%) | 0.033 | 1.40 | 1.03–1.92 | 439 (30.1%) | 41 (39.0%) | 0.054 | 1.49 | 0.99–2.24 |
| | 36–59 | 840 (61.4%) | 101 (51.5%) | 0.009 | 0.67 | 0.49–0.90 | 887 (60.8%) | 54 (51.4%) | 0.059 | 0.68 | 0.46–1.00 |
| | 60+ | 122 (8.9%) | 22 (11.2%) | 0.179 | 1.29 | 0.79–2.09 | 134 (9.2%) | 10 (9.5%) | 0.906 | 1.04 | 0.53–2.05 |
| Gender | Male | 650 (47.5%) | 108 (55.1%) | 0.027 | 1.31 | 1.00–1.70 | 700 (47.9%) | 58 (55.2%) | 0.149 | 1.31 | 0.90–1.91 |
| | Female | 719 (52.5%) | 88 (44.9%) | | | | 760 (52.1%) | 47 (44.8%) | | | |
| Scil Outcome | No SCIL | 248 (18.1%) | 43 (21.9%) | 0.217 | 0.86 | 0.60–1.25 | 363 (24.9%) | 28 (26.7%) | 0.341 | 0.90 | 0.58–1.42 |
| | SCIL | 1121 (81.9%) | 153 (78.1%) | | | | 1097 (75.1%) | 77 (73.3%) | | | |
| Scil Outcome | Scil > 19 | 637 (62.4%) | 56 (36.6%) | 0.000 | | | 665 (60.6%) | 28 (26.7%) | 0.000 | | |
| | Scil 16–19 (BIF) | 202 (19.8%) | 40 (26.1%) | | | | 225 (20.5%) | 17 (16.2%) | | | |
| | Scil ≤15 (MID) | 182 (17.6%) | 57 (37.3%) | | | | 207 (18/9%) | 32 (30.5%) | | | |
| Borderline Intellectual Functioning (BIF) | SCIL > 19 | 637 (62.4%) | 56 (36.6%) | 0.000 | 2.49 | 1.83–3.39 | 665 (60.6%) | 28 (36.4%) | 0.000 | 2.52 | 1.61–3.95 |
| | SCIL ≤ 19 | 384 (37.6%) | 97 (63.4%) | | | | 432 (39.4%) | 49 (63.6%) | | | |
| Mild Intellectual Disability (MID) | SCIL >15 | 839 (82.2%) | 96 (62.7%) | 0.000 | 2.74 | 1.90–3.94 | 890 (81.1%) | 45 (58.4%) | 0.000 | 3.06 | 1.89–4.93 |
| | SCIL ≤15 | 182 (17.8%) | 57 (37.3%) | | | | 207 (18.9%) | 32 (41.6%) | | | |
| Diagnosis | Anxiety | 205 (15.0%) | 20 (10.2%) | 0.043 | 0.65 | 0.39–1.05 | 215 (14.7%) | 10 (9.5%) | 0.142 | 0.61 | 0.31–1.18 |
| | Depression | 429 (31.3%) | 39 (19.9%) | 0.001 | 0.54 | 0.38–0.79 | 449 (30.8%) | 19 (18.1%) | 0.006 | 0.50 | 0.30–0.83 |
| | Bipolar | 122 (8.9%) | 30 (15.3%) | 0.005 | 1.85 | 1.20–2.84 | 135 (9.2%) | 17 (16.2%) | 0.020 | 1.89 | 1.10–3.28 |
| | Psychotic disorder | 223 (16.3%) | 32 (16.3%) | 0.529 | 1.00 | 0.67–1.50 | 282 (19.3%) | 48 (45.7%) | 0.762 | 0.92 | 0.53–1.59 |
| | Schizophrenia | 256 (18.7%) | 74 (37.8%) | 0.000 | 2.64 | 1.92–3.63 | 239 (16.4%) | 16 (15.2%) | 0.000 | 3.52 | 2.35–5.28 |
| | Developmental disorder | 173 (12.6%) | 33 (16.8%) | 0.104 | 1.40 | 0.93–2.10 | 173 (11.8%) | 33 (31.4%) | 0.725 | 1.12 | 0.63–1.95 |
| | Alcohol and drug abuse disorder | 171 (12.5%) | 45 (23.0%) | 0.000 | 2.09 | 1.44–3.02 | 188 (12.9%) | 28 (26.7%) | 0.000 | 2.46 | 1.56–2.89 |
| | Personality disorder | 570 (41.6%) | 73 (37.2%) | 0.2430.243 | 0.83 | 0.61–1.13 | 610 (41.8%) | 33 (31.4%) | 0.037 | 0.64 | 0.42–0.98 |
| | Low GAF | 437 (31.9%) | 102 (52.0%) | 0.000 | 2.32 | 1.71–3.16 | 488 (33.4%) | 51 (52.4%) | 0.003 | 1.83 | 1.23–2.74 |

[1] For calculating the OR, the other categories are applied as reference, e.g. 0–35 is compared to 'else'

15 and below (assumed MID) was associated with more aggression in general (OR = 2.74), as well as with more *physical* aggression (OR = 3.06) (Table 2).

**SOAS-R, SCIL score and patient characteristics, univariate analyses.** Gender showed no significant association between aggression in general or more *physical* aggression. Only middle age showed an inverse and significant association with aggression (OR = 0.67, p = 0.009). Diagnosis of bipolar disorder (OR = 1.85, p = 0.005), schizophrenia (OR = 2.64,

p<0.001), alcohol and drug abuse disorder (OR = 2.09, p<0.001) and a low GAF (OR = 2.32, P<0.001) were associated with an increased risk of aggression.

Schizophrenia (OR = 3.52, p<0.001), drug abuse disorder (OR = 2.46, p<0.001), and a low GAF (OR = 1.83, P<0.003) were associated with an increased risk of *physical* aggression. Only depressive disorders (OR = 0.54, p = 0.001) were associated with less aggression in general and less physical aggression (OR = 0.50, p = 0.006).

**Logistic regression.** The logistic regression analysis showed that patients who screened positive for BIF (OR = 2.00, p = 0.003) or MID (OR 2.89, p<0.001) were more at risk of showing aggressive incidents, as well as the patients with the diagnoses bipolar disorder (OR 3.07, p<0.001), schizophrenia (OR 2.75, p<0.001), and a low GAF (OR 1.72, p = 0.005). Logistic regression analysis with *physical* aggression as an outcome showed that patients with MID (OR 2.50, p<0.001), a bipolar disorder (OR 3.13, p = 0.007) or schizophrenia (OR 4.04, p<0.001) were more at risk of showing aggressive incidents.

**Poisson regression.** These findings were underlined by the Poisson regression of the number of *physical* aggression incidents per patient. This showed anxiety disorder ($\beta = 0.62$, p<0.001), bipolar disorder ($\beta = 1.63$, p<0.001), schizophrenia ($\beta = 1.12$, p<0.001), developmental disorder ($\beta$) = 0.69, p<0.001), drug abuse disorder ($\beta = 1.18$, p<0.001) and a SCIL of 15 and below (Assumed MID; $\beta = 0.61$, p<0.001) were all related to more incidents.

In short, screening positive for BIF and MID were both associated with significantly more aggression, and this association appears to be somewhat stronger for MID. Bipolar disorder, developmental disorders, schizophrenia and drug abuse disorders are associated with higher aggression rates (Table 3).

## Discussion

The current study indicates that patients with a (suspected) BIF or MID are more likely to display aggressive incidents than patients without ID. The odds ratios for aggression of any type and *physical* aggression toward others are 2.5 to 2.9, making it a highly significant finding. This is in line with other studies with people with ID showing that ID is associated with higher rates of aggression incidents [29, 30]. Eight percent of patients without an ID had been engaged in aggressive incidents, keeping with previous Dutch data from international studies [18]. However, in the BIF patient group, 20%, and in the MID patient group, 24% of patients showed aggressive incidents, roughly half of which were *physical* aggression. This represents a sizable contribution to the risk of being confronted with aggression in the wards.

Regarding the number of incidents per patient, about half of the patients involved in aggression incidents were involved in 2–5 incidents, and just a small group of patients were responsible for more than five incidents. This is in keeping with previous studies. For instance, Bowers et al. [6] found that on average, 45% of patients with violent behaviour were involved in more than one incident. In a study by Broderick et al. [31] in a Canadian multihospital state psychiatric system, just 1% of the study population participated in 28.7% of all violent assaults. Considering the current findings, it seems fair to conclude that earlier studies and reviews concerning aggression in mental health may have paid too little attention to the role of impairments in intellectual functioning as a potential determinant of aggressive behaviour. This was also recently concluded in Weltens et al.'s systematic review [32]. However, Tsiouris et al. [33] reported in a large sample of persons with an ID that "impulse control, mood dysregulation and perceived threat appear to underlie most of the aggressive behaviours reported" across various settings. Another study in a Forensic Psychiatric Hospital [34] also concluded that chronic violent behaviour was associated with cognitive impairment or brain damage. In a study by Verstegen, a clear association between impulsivity and aggression was found [35]. Our study

**Table 3. Multivariable association between predictors and aggression.**

| | Predictors | | Aggression incidents | | | | | Physical aggression | | | | |
|---|---|---|---|---|---|---|---|---|---|---|---|---|
| | | | B | SE | sig | Ex (b) | 95% CI Ex (b) | B | SE | sig | Ex (b) | 95% CI Ex (b) |
| Univariable associations | Male Gender[1] | | 0.046 | 0.197 | 0.815 | 1.05 | 0.71–1.54 | 0.235 | 0.267 | 0.378 | 1.26 | 0.75–2.13 |
| | Age[2] | < 35 | 0.324 | 0.341 | 0342 | 1.38 | 0.71–2.69 | 0.721 | 0.485 | 0.137 | 2.06 | 0.79–5.32 |
| | | 35–59 | -0.248 | 0.325 | 0.445 | 0.78 | 0.41–1.47 | 0.134 | 0.470 | 0.776 | 1.14 | 0.45–2.87 |
| | Anxiety disorder[3] | | 0.366 | 0.303 | 0.288 | 1.44 | 0.79–2.61 | 0.191 | 0.444 | 0.668 | 1.21 | 0.51–2.89 |
| | Depressive disorder | | 0.255 | 0.266 | 0.338 | 1.29 | 0.76–2.17 | 0.135 | 0.377 | 0.721 | 1.14 | 0.55–2.39 |
| | Bipolar disorder | | 1.218 | 0.305 | <0.001 | 3.38 | 1.86–6.15 | 1.158 | 0.402 | 0.004 | 3.18 | 1.45–6.99 |
| | Psychotic disorder | | - 0.367 | 0.286 | 0.200 | 0.69 | 0.39–1.21 | - 0.428 | 0.387 | 0.269 | 0.65 | 0.31–1.39 |
| | Schizophrenia | | 1.122 | 0.263 | <0.001 | 3.07 | 1.84–5.14 | 1.337 | 0.338 | <0.001 | 3.81 | 1.96–7.39 |
| | Developmental disorder | | 0.735 | 0.276 | 0.008 | 2.08 | 1.21–3.58 | 0.475 | 0.386 | 0.218 | 1.61 | 0.75–3.43 |
| | Personality disorder | | -0.004 | 0.210 | 0.983 | 0.99 | 0.66–1.50 | - 0.481 | 0.300 | 0.109 | 0.62 | 0.34–1.11 |
| | Alcohol and drug abuse disorder | | 0.643 | 0.240 | 0.007 | 1.90 | 1.19–3.05 | 0.703 | 0.308 | 0.022 | 2.02 | 1.11–3.69 |
| | Low GAF | | 0.596 | 0.194 | 0.002 | 1.81 | 1.24–2.68 | 0.264 | 0.260 | 0.311 | 1.30 | 0.78–2.17 |
| | SCIL[4] | 16–19 (BIF) | 0.716 | 0.239 | 0.002 | 2.07 | 1.30–3.31 | 0.379 | 0.333 | 0.254 | 1.46 | 0.76–2.80 |
| | | ≤15 (MID) | 1.115 | 0.228 | <0.001 | 3.05 | 1.96–4.75 | 0.986 | 0.295 | <0.001 | 2.68 | 1.50–4.78 |
| Final model | Age | <35 | 0.562 | 0.199 | 0.005 | 1.75 | 1.18–2.59 | 0.632 | 0.257 | 0.014 | 1.88 | 1.14–3.11 |
| | SCIL score between 16 and 19 (BIF) | | 0.684 | 0.237 | 0.004 | 1.98 | 1.25–3.15 | | | | | |
| | SCIL score 15 and below (MID) | | 1.080 | 0.223 | <0.001 | 2.94 | 1.90–4.56 | 0.850 | 0.257 | <0.001 | 2.34 | 1.41–3.88 |
| | Bipolar disorder | | 1.075 | 0.279 | <0.001 | 2.93 | 1.69–5.05 | 1.102 | 0.364 | 0.001 | 3.22 | 1.58–6.58 |
| | Psychotic disorder | | -0.463 | 0.273 | 0.090 | 0.63 | 0.37–1.07 | | | | | |
| | Schizophrenia | | 0.954 | 0.219 | <0.001 | 2.59 | 1.69–3.99 | 1.370 | 0.268 | <0.001 | 3.93 | 2.33–6.65 |
| | Developmental disorder | | 0.616 | 0.261 | 0.018 | 1.85 | 1.10–3.09 | | | | | |
| | Alcohol and drug abuse disorder | | 0.560 | 0.231 | 0.015 | 1.75 | 1.11–2.75 | 0.638 | 0.286 | 0.026 | 1.89 | 1.08–3.32 |
| | Low GAF | | 0.602 | 0.194 | 0.002 | 1.83 | 1.25–2.67 | | | | | |

[1] Female gender was used as reference category

[2] Elderly was used as reference category.

[3] No diagnosis was used as reference category

[4] SCIL > 19 was used as reference category.

confirmed that BIF/MID is an often unnoticed and undiagnosed factor that significantly contributes to *physica*lly aggressive behaviour, supporting our knowledge that patients with lower cognitive functioning, in general, may have more problems with impulse regulation.

Our findings suggest that specific patient characteristics increase the risk of being involved in aggressive incidents. These include young ages of up to 35 years, which is in line with other studies with inpatients and reviews in adult psychiatry and ID populations [6, 29]. Other characteristics that showed an increased aggression risk were diagnosis of schizophrenia, harmful use of alcohol and drugs and bipolar affective disorder, and a GAF score below 45. The same patient characteristics were important when analysing only *physical* aggression incidents, showing that aggressive incidents per se appear to have similar patient risk factors. While this is one of the first comprehensive studies examining the association between ID and risk of aggression, schizophrenia and drug and alcohol use have commonly been associated with aggression in mental health care [6, 34, 35] and studies with people with ID. [12, 16, 29].

As we can learn from studies with people with ID based on interviews with people with intellectual disability [10, 36], people with ID often experience a lack of structure in their daily life, and staff may often place too many demands on them. This is coupled with the fact that these patients regularly have difficulty dealing with emotions, the complexity of social interaction, and other stressors. Challenging behaviour can also be related to a number of unmet needs that should be addressed, such as medical issues (e.g., pain) or communication difficulties, among others. It is also important to better understand patients' capabilities by staff and others—both in terms of their intellectual, emotional and adaptive skills. It may be helpful to offer patients counselling or training to better cope with emotions and impulses to help to reduce the occurrence of CB. Other potential ways forward are functional analyses of earlier CB and positive behaviour support for patients. The results of a large-scale meta-analysis indicated that in patients/clients with ID, behavioural treatments based on Function Analysis tend to be more effective than pharmacological interventions [37]. Function Analysis derived from Behaviour Therapy to systematically identify the reinforcers of CB allows staff to mitigate the consequence and replace it with more prosocial behaviour [13, 38–41].

## Clinical implications

Earlier studies examining the associations between aggressive behaviour and patient characteristics focused on diagnoses, psychiatric history, staff training, restraint and workload, the interaction between patient and staff, care processes, and ward architecture and environment [31, 42]. Based on the information from these studies, various suggestions and programs to reduce aggression were developed. We know from studies with people with ID that it is advisable to meet a person's needs better (e.g., engaging in appropriate support and communication, presenting information in an accessible way, and approaching treatment from a biopsychosocial approach), all of which would likely result in fewer behaviours that challenge [43–46]. Until now, mental health services seem to be frequently not equipped enough to meet those needs. The current study reminds us of the importance of the intellectual functioning of each individual patient in general psychiatry to prevent aggressive incidents, especially those responsible for a large number of incidents. In light of our findings, we recommend screening patients for ID as part of any assessment at the start of treatment in order to support the prevention of aggressive incidents in psychiatric care. The SCIL can be helpful in psychiatric care [28, 47] and give a quick first impression. Treatment and support to the needs can be adapted according to the SCIL category. As such, avoid over-demanding and stress. The staff's attitudes towards people with ID may also play a part in detecting and preventing aggressive incidents [16, 48].

## Recommendations

In mental health care, future studies should examine the causes and reasons for aggressive incidents in patients with MID/BIF and learn more about how these patients differ or resemble those in other studies with people/patients with ID. For example, we do not yet know much about the mental problems in the patient group with ID in mental health care and how this affects behaviour. We know from another of our studies [2] that patients with assumed ID frequently experience neglect and (sexual)abuse, but this is not recognised enough in mental health care. We also do not know the influence of social factors between patients within ward settings, as the current study did not look into that level of detail.

Next, intervention studies in patients with MID and BIF are needed in mental health care to study which measures (such as positive behaviour support [48, 49]) we know are helpful in ID care and preventing aggressive incidents. Cooperation with ID services can therefore be

helpful. Attention to this vulnerable, large patient group in mental health is of great importance and should get more attention in the training of professionals.

## Limitations and strengths

In this study, we included outpatients and inpatients over a period of six years. This gave us a large sample of consecutive patients in a quite big mental healthcare trust. This large sample also provided the opportunity to study predictors of incidents of *physical* aggression as a subgroup of all aggression. Another strength is the high percentage of included patients, resulting in a representative sample. A limitation is that aggression incidents are likely to have been underreported with the SOAS-R.

## Supporting information

**S1 Data.**
(XLSX)

**S2 Data.**
(XLSX)

**S1 File.**
(DOCX)

## Acknowledgments

Sadly, Prof. Henk Nijman passed away during the writing process. We want to express our gratitude and respect to an outstanding person and researcher.

## Author Contributions

**Conceptualization:** Jeanet Grietje Nieuwenhuis, Cornelis Lambert Mulder, Henk Liewellyn Inge Nijman, Eric Onno Noorthoorn.

**Data curation:** Eric Onno Noorthoorn.

**Formal analysis:** Jeanet Grietje Nieuwenhuis, Cornelis Lambert Mulder, Henk Liewellyn Inge Nijman, Eric Onno Noorthoorn.

**Investigation:** Jeanet Grietje Nieuwenhuis.

**Methodology:** Jeanet Grietje Nieuwenhuis, Peter Lepping, Eric Onno Noorthoorn.

**Supervision:** Eric Onno Noorthoorn.

**Writing – original draft:** Jeanet Grietje Nieuwenhuis, Eric Onno Noorthoorn.

**Writing – review & editing:** Peter Lepping, Cornelis Lambert Mulder, Henk Liewellyn Inge Nijman.

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
