## [Decision Letter · Decision Letter 0]

21 Mar 2022

PONE-D-21-40785Aggressive behaviour of psychiatric patients with Mild and Borderline Intellectual Disabilities in general Mental Health CarePLOS ONE

Dear Dr. Nieuwenhuis,

Thank you for submitting your manuscript to PLOS ONE. After careful consideration, we feel that it has merit but does not fully meet PLOS ONE’s publication criteria as it currently stands. Therefore, we invite you to submit a revised version of the manuscript that addresses the points raised during the review process. I think your paper is important and of clinical interest and I would therefore consider your paper for publication pending some revisions. You should have no problems in addressing the comments as these are mainly on clarifications and explanations regarding statistics (e.g., missings), instruments (e.g., cut off values) and procedures used. Please consider using people-first language as reviewer #1 noted in his comments. 

We look forward to receiving your revised manuscript.

Kind regards,

Robert Didden

Academic Editor

PLOS ONE

Journal Requirements:

Reviewers' comments:

Reviewer's Responses to Questions

**Comments to the Author**

1. Is the manuscript technically sound, and do the data support the conclusions?

Reviewer #1: No

Reviewer #2: Yes

2. Has the statistical analysis been performed appropriately and rigorously? 

Reviewer #1: No

Reviewer #2: Yes

3. Have the authors made all data underlying the findings in their manuscript fully available?

Reviewer #1: Yes

Reviewer #2: Yes

4. Is the manuscript presented in an intelligible fashion and written in standard English?

Reviewer #1: Yes

Reviewer #2: Yes

5. Review Comments to the Author

Reviewer #1: Thank you for the opportunity to review this manuscript, which examines the frequency of aggressive behaviours among people with mild intellectual disability/borderline intellectual functioning within a general mental health care setting. The large sample size is a strength of this study, but I have concerns about the novelty of the findings, the lack of depth of discussion of factors associated with behaviours that challenge, and the language used throughout. People with intellectual disability and co-occurring mental illness have multiple and complex support needs, and yet health and mental health service systems are frequently not equipped to meet those needs. It is already known that people with intellectual disability are more likely than people without intellectual disability to have behaviours that challenge including aggression towards self and others. The factors contributing to such behaviours can be complex, including difficulties with communication, stressors within the environment, and underlying physical and mental health conditions (e.g. physical pain, mental illness), among others. These factors can be proactively addressed but currently the manuscript lacks any discussion about how to equip mental health services to do so. I would also very much recommend that the authors review the language used throughout this manuscript. I have made some specific comments below.

Introduction

• It would be appropriate to use person-first language throughout, i.e. rather than “ID patients” say people with ID

• There is inconsistent use of ID and intellectual disability throughout

• Intellectual disability does not need to be capitalized, same with bipolar disorder

• Page 3: Plagued is a very emotive word and I would not consider it appropriate in this context. Suggest just saying that most previous studies have been single centre reports and say why this is a limitation.

Method

• Page 4: 630.000 should read 630,000

• Page 4: can outpatient psychiatric clinics be defined, as done for other types of care

• Page 4: expand SMI on first mention

• Page 6: expand GAF

• Page 6: more information is needed for the logistic regression analysis, what were the outcomes? What were the predictors? Were these included in a single model?

Results

• Page 7: How was missing SCIL data handled? E.g. in the SOAS-R data the proportion is reported with 1174 as the denominator (196/1174=16.7%), but for others the complete sample is used (36-1565=2.3%)

• Similar to above, suggest checking the rounding of decimal places as some of these are rounded incorrectly assuming a total sample of 1565

• Page 7, “to” is missing between “84 patients in two” and “up to five incidents”

• Figures 1 & 2: These appear to be cumulative frequencies and are not an appropriate or helpful way to present these data. This information is adequately described in text without figures.

• Table 2: Reference categories are not indicated and in some cases not provided (e.g. age categories). Why is age and SCIL included as both a continuous and categorical variable? I am also unsure of the appropriateness of having two separate variables for SCIL score (i.e. >19 vs ≤19 and then >15 vs ≤15); why not have a single variable with exclusive categories for people with no mild intellectual disability or borderline intellectual functioning, people with borderline intellectual functioning, and people with mild intellectual disability (SCIL scores >19, 16-≤19, and ≤15, respectively?)

• Table 3: reference categories are not provided but are needed for interpretation. Why do categories for things like age and SCIL change to those provided in Table 2? Why are only significant results presented?

• Page 8: did the authors perform a linear regression with the number of physical aggression incidents as the outcome? (“These findings were underlined by the regression of the number of physical aggression incidents per patient, which detected bipolar disorder (β=0.169, p=0.014), schizophrenia (β=0.144, p0.008), drug abuse disorder (β=0.195, p0.001) and a SCIL below 15 (β=0.138, p=0.009) as predictors.”) If so this is the first mention of it; this should have been described in the analysis section.

Discussion:

• I have concerns about the overall tone and language used throughout the manuscript but particularly the discussion. Terms such as “aggressive patients” and “violent patients” are stigmatising and not helpful. Similarly, describing people with intellectual disability as having an “inability to properly cope with emotions” is also not helpful, especially when considering the complex health and support needs of people with intellectual disability and co-occurring mental illness, and the systemic neglect and abuse people with intellectual disability frequently experience within health and mental health systems.

• Similar to the introduction, people first language is more appropriate- for example instead of saying “ID studies based on patient interviews”, say interviews with people with intellectual disability.

• The authors draw a number of associations in their discussion but on little basis – e.g. using their data as evidence of problems with impulse regulation; behaviours that challenge can also be related to a number of unmet needs that should be addressed such medical issues (e.g. pain), communication difficulties etc.

• Topics such as positive behviour support are mentioned only briefly in the discussion, yet this information is much more helpful for clinicians for preventing behaviours than other text provided in the discussion

• In the conclusion and clinical implications it would be more appropriate and helpful to state that screening for intellectual disability in a mental health setting may be helpful to inform strategies to better meet a person’s needs (e.g. engaging appropriate supports, presenting information in an accessible way, approaching treatment from a biopsychosocial approach, all of which would likely result in fewer behaviours that challenge).

Reviewer #2: This is interesting research that explores the frequency of aggressive behaviours and incidents among people suspected to have MID/BIF.

Introduction

- Are there any statistics that show the prevalence use of coercive or restrictive measures in Netherlands on people with MID and BIF?

- Explanation about the rationale for the aims of the research study is recommended. How would it useful for clinicians when they identified patients suspected to have MID/BIF in their work?

- Explanation about the use of screening tools to screen suspected patients with MID/BIF instead of using proper diagnosis as defined by ICD/DSM is recommended.

Methods

- Lack of details regarding the procedures. Does the SOAS-R administered by the same staff for the patients or administered by different staffs taking care of the patient?

- How does the screening of the participants using SCIL carried out?

- Does the staff administer both SCIL and SOAS-R for the same patient?

- Do the patients have both MID/BIF and co-morbid mental health conditions? How would the results take into account that the patient may display aggressive behaviours not because of their cognitive functioning and rather their mental health conditions?

- In the results table (Table 2 and 3) – how is developmental disorder defined? Some developmental disorders may include symptoms of MID/BIF, how will that affect the results?

Discussion

- Some patients are staying in long stay wards – does social factors play a part that may contribute to the aggressive behaviours instead of cognitive impairment

- How does screening patients for ID support the prevention of aggressive incidents as mentioned in clinical implications?

- Do the staff’s attitudes towards people with ID play a part to detect and prevent aggressive incidents?

- As the staff interacts with the patients for a long period, it is a possibility that their relationship with the patients may influence their ratings on the scales, would it become discriminatory tool to segregate people with suspected ID and those without?

6. PLOS authors have the option to publish the peer review history of their article (what does this mean?). If published, this will include your full peer review and any attached files.

Reviewer #1: No

Reviewer #2: No

---

## [Author Response · Author response to Decision Letter 0]

20 May 2022

PONE-D-21-40785

Response to the reviewers' comments. We apply italics to show the reviewers questions and plain text for our answers:

Reviewer #1: 

Question # 1 Thank you for the opportunity to review this manuscript, which examines the frequency of aggressive behaviours among people with mild intellectual disability/borderline intellectual functioning within a general mental health care setting. The large sample size is a strength of this study, but I have concerns about the novelty of the findings, the lack of depth of discussion of factors associated with behaviours that challenge, and the language used throughout. 

Response #1

We thank the reviewer for this comment. Indeed, this study is one of the largest naturalistic empirical studies we know of. With respect to the concern of the novelty of the findings, in the discussion (page 10 line 316) we refer to Weltens et al of October 2021. They noted in a review of aggression factors on admission wards in Mental Health Care, that an IQ of 50 - 84 is a risk factor for coercive measures. They noted that this has not been studied as a direct factor for aggression before. They suggest that it should be a topic for future research. In our point of view, this is the main topic of the current study, which according to Weltens, has not been studied in detail before. 

Question # 2. People with intellectual disability and co-occurring mental illness have multiple and complex support needs, and yet health and mental health service systems are frequently not equipped to meet those needs. It is already known that people with intellectual disability are more likely than people without intellectual disability to have behaviours that challenge, including aggression towards self and others. 

The factors contributing to such behaviours can be complex, including difficulties with communication, stressors within the environment, and underlying physical and mental health conditions (e.g., physical pain, mental illness), among others. These factors can be proactively addressed, but currently the manuscript lacks any discussion about how to equip mental health services to do so. I would also very much recommend that the authors review the language used throughout this manuscript. I have made some specific comments below.

Response #2 Discussion: Thank you for this comment. We added text to the current draft to cover these discussion points (page 11, line 334-347): 

As we can learn from ID studies based on interviews with people with intellectual disability [10, 37], people with ID often experience a lack of structure in their daily life, and staff may often be placing too many demands on them. This is coupled with the fact that these patients regularly have difficulty in dealing with emotions, the complexity of social interaction, and other stressors. Behaviours that challenge can also be related to a number of unmet needs that should be addressed such medical issues (e.g., pain), communication difficulties etc. It is also important to have a better understanding of patients' capabilities by staff and others - both in terms of their intellectual, emotional and adaptive skills. It may bel be helpful offering counselling or training to better cope with emotions and impulses to help to reduce the occurrence of CB. Other potential ways forward are functional analyses of earlier CB and positive behaviour support for patients. The results of a large-scale meta-analysis indicated that in clients with ID, behavioural treatments based on Function Analysis tend to be more effective than pharmacological interventions [38]. Function Analysis derived from Behaviour Therapy to systematically identify the reinforcers of CB allows staff to mitigate the consequence and replace it with more prosocial behaviour [13, 40-42]. 

Introduction  

Question #3• It would be appropriate to use person-first language throughout, i.e., rather than “ID patients” say people with ID; 

Response # 3 This is done.

Question #4 There is inconsistent use of ID and intellectual disability throughout; 

Response #4 This is done. We used intellectual disability at first mention and ID throughout. 

Question #5 Intellectual disability does not need to be capitalized, same with bipolar disorder. 

Response#5 We adjusted this. As stated above we used the full word (intellectual disability) at first mention and the abbreviation (ID) later on. 

Question #6 Page 3: Plagued is a very emotive word and I would not consider it appropriate in this context. Suggest just saying that most previous studies have been single centre reports and say why this is a limitation. 

Response # 6/ We changed this, page 4, line 93: Many previous publications have been single centre reports, which makes comparison and generalizing conclusions difficult. 

Method

Question #7 Page 4: 630.000 should read 630,000; 

Response #7. This was changed, page 4, line 129.

Question #8 Page 4: can outpatient psychiatric clinics be defined, as done for other types of care; 

Response #8. We explained this as follows (page 5, line 131): 1. Outpatient psychiatric clinics in this context are the services patients are referred to by the General Practitioner for initial mental health care. This service provides acute crisis interventions as well as outpatient psychological and psychiatric treatment and support. 

We also refined the next sentence (page 5 line 134): 2. Flexible Assertive Community Treatment (FACT) teams specialised in daily (outpatient) support and treatment for Serious Mentally Ill (SMI) patients. In the Netherlands, FACT teams are multidisciplinary outpatient teams with 8–10 professionals, such as a psychiatrist, psychologist, nurses, and social workers, generally caring for 200 SMI patients. 

Question #9 Page 4: expand SMI on first mention; 

Response #9 Page 5 line 135: Serious Mentally Illness (SMI) 

Question #10 Page 6: expand GAF; 

Response #10 this was done page 6 line 192

Question # 11 Page 6: more information is needed for the logistic regression analysis, what were the outcomes? What were the predictors? Were these included in a single model? 

Response # 11 We added the following section to the statistical analysis, page 6 line 207 onwards: 

A forward entry, backward deselection procedure was used. All variables selected from the EMC were entered in the analysis, thus gender, age categories, diagnosis, MID or BIF as assessed with the SCIL. For the forward selection, variables with associations having a p-value of <0.2 were included in the logistic regression analysis, following the relevance criterion proposed by Hosmer and Lemeshow [26]. These were entered in 3 blocks, first the demographic variables, then the diagnoses and finally the response categories in the SCIL. 

Next, a Poisson regression was applied to the number of incidents as we may expect a skewed distribution and the number of incidents represents a count. Before applying the regression, the distribution of the number of incidents was tested. We applied forward entry, backward deselection to investigate which patient characteristics predicted the number of aggression incidents. We present the Ex(β), which can be interpreted as a growth or downturn rate [27]. 

Results

Question # 12 Page 7: How was missing SCIL data handled? E.g., in the SOAS-R data the proportion is reported with 1174 as the denominator (196/1174=16.7%), but for others the complete sample is used (36-1565=2.3%). 

Response #12, We thank the reviewer for this comment. We understand the confusion regarding the mentioning of various numbers in the first paragraphs of the results section. To clarify this, we first presented the aggression incidents in the full sample. We then present the number of SCIL’s assessed. We did not handle missing data, but described it in more detail in table 2. We did this deliberately to allow some insight into those patients without a SCIL outcome, as compared to those with a SCIL outcome. Especially to answer the question whether aggression occurs more in patients not assessed with the SCIL than in those without. This proved not to be the case, as can be seen in table 2, line 10, where we compared them. Please note that table 2 had the full sample as denominator. Table 1 shows the findings from all patients who have had a SCIL. We chose to describe what was assessed. 

Question # 13 Similar to above, suggest checking the rounding of decimal places as some of these are rounded incorrectly assuming a total sample of 1565. 

Response #13. We checked all tables, with respect to counters and rounding of percentages. 

Question # 14 Page 7, “to” is missing between “84 patients in two” and “up to five incidents”. 

Response #14. We corrected this. 

Question # 15 Figures 1 & 2: These appear to be cumulative frequencies and are not an appropriate or helpful way to present these data. This information is adequately described in text without figures.

Response #15. We removed the figures and extended the text page 7, line 229 onwards: 

 In total, we found 1472 aggressive incidents in 196 (16.7%) of the 1565 patients. Most of the registered incidents occurred in inpatients. Only 36 outpatients were involved (18.3 % of the 196, 2.2% of the complete sample). Of the 196 patients with an incident of aggression, 47 were involved in one incident, 84 patients between two and five incidents, and 65 patients were involved in over six incidents. 23 (11.7% of 196) patients were responsible for 751 aggression incidents (51.0% of 1472). The mean number of incidents was 7.53 per patient, with a maximum of 78 incidents. Of the 1565 patients, 105 were engaged in 269 physical, outwardly aggressive incidents (18.3% of the 1472 incidents). Of these 105 patients, 46 were involved in one incident, 51 in between two and five incidents, and 8 in over six physically aggressive incidents. 20 (7.4%) of these patients were responsible for 137 (50.9%) of the 269 incidents. Both analyses show that approximately 10% of the patients account for half of the aggression incidents.

The current study reminds us of the importance of the intellectual functioning of each individual patient in general psychiatry in an attempt to prevent aggressive incidents, especially those responsible for a large number of incidents. 

Question # 16 Table 2: Reference categories are not indicated and, in some cases, not provided (e.g., age categories). Why is age and SCIL included as both a continuous and categorical variable? I am also unsure of the appropriateness of having two separate variables for SCIL score (i.e., >19 vs ≤19 and then >15 vs ≤15); why not have a single variable with exclusive categories for people with no mild intellectual disability or borderline intellectual functioning, people with borderline intellectual functioning, and people with mild intellectual disability (SCIL scores >19, 16-≤19, and ≤15, respectively?). 

Response #16. We chose to analyze the categories in the SCIL, identifying BIF and MID as these were developed to assist treatment decision processes (Seelen et al, 2019). As suggested we first present them in three groups (SCIL scores >19, 16-≤19, and ≤15) and then in two (19 vs ≤19 and then >15 vs ≤15). The last was done to calculate odds ratios.

Seelen-de Lang BL, Smits HJH, Penterman BJM, Noorthoorn EO, Nieuwenhuis JG, Nijman HLI. Screening for intellectual disabilities and borderline intelligence in Dutch outpatients with severe mental illness. J Appl Res Intellect Disabil. 2019 Sep;32(5):1096-1102. doi: 10.1111/jar.12599. Epub 2019 Apr 29. 

Question # 17. Table 3: reference categories are not provided but are needed for interpretation. Why do categories for things like age and SCIL change to those provided in Table 2? Why are only significant results presented? 

Response # 17. A forward entry and backward deselection procedure were used, and the final model was presented. We added the uncorrected univariable comparisons to the table to improve interpretation. The reference categories can now be checked in the new table 3, as they are noted below the table. 

Question # 18. Page 8: did the authors perform a linear regression with the number of physical aggression incidents as the outcome? (“These findings were underlined by the regression of the number of physical aggression incidents per patient, which detected bipolar disorder (β=0.169, p=0.014), schizophrenia (β=0.144, p0.008), drug abuse disorder (β=0.195, p0.001) and a SCIL below 15 (β=0.138, p=0.009) as predictors.”) If so, this is the first mention of it; this should have been described in the analysis section. 

Response # 18. No, we performed a Poisson regression analysis of the number of aggression incidents. We elaborated on this analysis on page 7 line 214 onwards, as follows: 

Next, a Poisson regression was applied to the number of incidents as we may expect a skewed distribution and the number of incidents represents a count. Before applying the regression, the distribution of the number of incidents was tested. We applied forward entry, backward deselection to investigate which patient characteristics predicted the number of aggression incidents. We present the Ex(β), which can be interpreted as a growth or downturn rate [27]. 

Also in the results section, page 9 line 287 onwards: 

Poisson regression 

 These findings were underlined by the regression of the number of physical aggression incidents per patient, which detected the following as predictors: anxiety disorder (Ex(β)=2.01, p<0.001), bipolar disorder (Ex(β)=5.14, p<0.001), schizophrenia (Ex(β)=3.11, p<0.001), developmental disorder (Ex(β)=2.11, p<0.001), drug abuse disorder (β=3.21, p0.001), and a SCIL below 15 (Ex(β)=1.74, p<0.001). 

Discussion: 

Question # 19 I have concerns about the overall tone and language used throughout the manuscript but particularly in the discussion. Terms such as “aggressive patients” and “violent patients” are stigmatising and not helpful; 

Response # 19. We changed this to: involved in aggression incidents in lines 84, 308, 311 and in line 325 to patients with violent behaviour as the articles refer to violence. 

Question # 20. Similarly, describing people with intellectual disability as having an “inability to properly cope with emotions” is also not helpful, especially when considering the complex health and support needs of people with intellectual disability and co-occurring mental illness, and the systemic neglect and abuse people with intellectual disability frequently experience within health and mental health systems;

Response 20 # We changed this to: these patients regularly have difficulty in dealing with emotions in line and mentioned the complex need in the discussion (lines 336 -341), as follows: 

This is coupled with the fact that these patients regularly have difficulty in dealing with emotions, the complexity of social interaction, and other stressors. Behaviours that challenge can also be related to a number of unmet needs that should be addressed such medical issues (e.g., pain), communication difficulties etc. It is also important to have a better understanding of patients' capabilities by staff and others - both in terms of their intellectual, emotional and adaptive skills. It may bel be helpful offering counselling or training to better cope with emotions and impulses to help to reduce the occurrence of CB. 

Question # 21 Similar to the introduction, people first language is more appropriate- for example instead of saying “ID studies based on patient interviews”, say interviews with people with intellectual disability; 

Response #21 We changed in line 334 as follows: As we can learn from ID studies based on interviews with people with intellectual disability 

Question #22. The authors draw a number of associations in their discussion but on little basis – e.g., using their data as evidence of problems with impulse regulation.

Response #22 We now used the words, eg in line 325: 

Our findings suggest that specific patient characteristics increase the risk of being involved in aggressive incidents. 

Later in the same paragraph we say that these findings are in line with other studies in psychiatry and in ID studies, line 331: 

While this is one of the first comprehensive studies examining the association between ID and risk of aggression, schizophrenia and drug and alcohol use have commonly been associated with aggression in mental health care [ 6, 35, 36] and ID studies [12 16, 30]. 

We therefore think that we do not draw such conclusions from our own study.

Question # 23 Behaviours that challenge can also be related to a number of unmet needs that should be addressed such medical issues (e.g. pain), communication difficulties etc. 

Response #23a We added this in line 336: Behaviours that challenge can also be related to a number of unmet needs that should be addressed such medical issues (e.g., pain) or communication difficulties among others. 

Response #23b Next, we added in clinical implications, lines 354-357: We know from studies in people with ID that it is advisable to better meet a person’s needs with engaging appropriate supports, presenting information in an accessible way, approaching treatment from a biopsychosocial approach, all of which would likely result in fewer behaviours that challenge. Until now, mental health services frequently do not seem to be adequately equipped to meet those needs.

Response #23c In recommendations, we added some lines (368-372) for future research: 

In mental health care, future studies should examine the causes and reasons are for aggression incidents in patients with MID/BIF, and to learn more about how these patients differ or resemble those from people in other ID studies. For example, we do not yet know much about the mental problems in the ID patient group in MHC, and how this affects behaviour. We know from another of our studies [2] that patients with ID frequently experience neglect and (sexual) abuse , but this is not necessarily recognized enough in mental health care. We also do not know the influence of social factors between patients within ward settings, as the current study did not look into that level of detail. 

Question # 24 Topics such as positive behaviour support are mentioned only briefly in the discussion, yet this information is much more helpful for clinicians for preventing behaviours than other text provided in the discussion.

Response #24 We added several lines in the discussion and clinical implications, such as lines 354 – 357: 

We know from studies in people with ID that, it is advisable to better meet a person’s needs (e.g., engaging appropriate supports and communication, presenting information in an accessible way, approaching treatment from a biopsychosocial approach), all of which would likely result in fewer behaviours that challenge. Until now mental health services systems seem to be frequently not equipped enough to meet those needs. 

And lines 375- 379:

Next, intervention studies in patients with MID and BIF are needed in mental health care to study which measures (such as positive behaviour support) that we know are helpful in ID care, are effective in preventing aggressive incidents. Cooperation with ID services can therefore be useful. Attention to this vulnerable, large patient group in the mental health is of great importance and should get more attention in the training of professionals.

Question # 25 In the conclusion and clinical implications, it would be more appropriate and helpful to state that screening for intellectual disability in a mental health setting may be helpful to inform strategies to better meet a person’s needs (e.g., engaging appropriate supports, presenting information in an accessible way, approaching treatment from a biopsychosocial approach, all of which would likely result in fewer behaviours that challenge).

Response #25 This was addressed in line 358-365: The current study reminds us of the importance of the intellectual functioning of each individual patient in general psychiatry in an attempt to prevent aggressive incidents, especially those responsible for a large number of incidents. In light of our findings, we recommend screening patients for ID as part of any assessment at the start of treatment in order to support the prevention of aggressive incidents in psychiatric care. The SCIL can be helpful in psychiatric care [29,45] and give a quick first impression. Treatment and support to the needs can then be adapted according to the SCIL category. As such, avoid over-demanding and stress. The staff’s attitudes towards people with ID may also play a part to detect and prevent aggressive incidents. 

 

Reviewer #2: This is interesting research that explores the frequency of aggressive behaviours and incidents among people suspected to have MID/BIF.

Introduction

Question # 26 Are there any statistics that show the prevalence use of coercive or restrictive measures in Netherlands on people with MID and BIF?

Response # 26. Yes, one of the first studies of the same research line investigated this. In the study of Nieuwenhuis et al, 2017 these Figures were presented: Nieuwenhuis JG, Noorthoorn EO, Nijman HL, Naarding P, Mulder CL. A Blind Spot? Screening for Mild Intellectual Disability and Borderline Intellectual Functioning in Admitted Psychiatric Patients: Prevalence and Associations with Coercive Measures. PLoS One. 2017 Feb 2;12(2):e0168847. doi: 10.1371/journal.pone.0168847. 

Also, a nationwide study confirmed this association: 22. Noorthoorn EO, Voskes Y, Janssen WA, Mulder CL, van de Sande R, Nijman HL, Smit A, Hoogendoorn AW, Bousardt A, Widdershoven GA (2016). Seclusion Reduction in Dutch Mental Health Care: Did hospitals meet goals set? Psychiatric Services, 1;67(12):1321-1327 DOI: appips201500414.

We added these references to the reference list and included the following text in the introduction, page 4, line 109: 

In a study on admission wards [1], we showed patients with BIF/ MID had an increased risk of involuntary admission (OR 2.71; SD 1.28-5.70) and coercive measures (OR 3.95, SD 1.47-10.54). These findings were confirmed in nationwide data gathered in 2014, where intellectual impairment also showed an association with increased risk of seclusion and other coercive measures [22]. Internationally, there is evidence that patients with BIF/ID account for more and longer seclusion and restraint events [2,3]. 

Thus, the following references were added: 

2. Turner KV & Mooney P A comparison of seclusion rates between intellectual disability and non-intellectual disability services: the effect of gender and diagnosis, The Journal of Forensic Psychiatry & Psychology, 2016, 27:2, 265-280, Doi 10.1080/14789949.2015. 1122822

3. Lepping P, Masood B, Flammer E, Noorthoorn EO. Comparison of restraint data from four countries. Soc Psychiatry Psychiatr Epidemiology. 2016 Sep;51(9):1301-9. doi: 10.1007/s00127-016-1203-x. 

Question # 27. Explanation of the rationale for the aims of the research study is recommended. How would it useful for clinicians when they identified patients suspected to have MID/BIF in their work? 

Response # 27 This was addressed in the lines 357-361: The current study reminds us of the importance of the intellectual functioning of each individual patient in general psychiatry in an attempt to prevent aggressive incidents, especially those responsible for a large number of incidents. In light of our findings, we recommend screening patients for ID as part of any assessment at the start of treatment in order to support the prevention of aggressive incidents in psychiatric care. 

Question # 28. Explanation about the use of screening tools to screen suspected patients with MID/BIF instead of using proper diagnosis as defined by ICD/DSM is recommended. 

Response #28. We added the following lines (362-366): The SCIL can be helpful in psychiatric care [29,45] and give a quick first impression. Treatment and support to the needs can then be adapted according to the SCIL category. As such, avoid over-demanding and stress. The staff’s attitudes towards people with ID may also play a part to detect and prevent aggressive incidents. 

Also, the suggestion of the reviewer addresses an issue which is the motivation to perform a line of studies into MID and BIF. From these former studies, we identified an underreport in the ‘proper diagnoses’. In our point of view clinicians do not estimate the intellectual abilities of patients completely adequate, especially in the serious mentally ill patients. This was especially addressed in the publication : Nieuwenhuis JG, Lepping P, Mulder NL, Nijman HLI, Veereschild M, Noorthoorn EO. Increased prevalence of intellectual disabilities in higher-intensity mental healthcare settings. BJPsych Open. 2021 Apr 22;7(3):e83. doi: 10.1192/bjo.2021. 28. In the introduction this is addressed in the first sentences (page 3 line 63 - 67: 

Mild Intellectual Disability (MID) and Borderline Intellectual Functioning (BIF) are highly prevalent in general Mental Health Care but often stay unnoticed [1,2]. Our research group has previously shown that in the Netherlands, the prevalence of MID/BIF increases by setting, from 27% in outpatient settings, to 40% in Flexible Assertive Community Treatment (FACT) teams and admission wards, to 67% in long-stay wards [3]. 

Methods

Question #29 Lack of details regarding the procedures. Does the SOAS-R administered by the same staff for the patients or administered by different staff taking care of the patient? How does the screening of the participants using SCIL carried out? Does the staff administer both SCIL and SOAS-R for the same patient? 

Response # 29 These three questions were addressed in page 6, line 185-189: 

Nurses working both in inpatient and outpatient settings were trained to administer the SCIL. According to the SCIL questionnaire instructions, the SCIL was administered by a person not involved in the treatment. In the Mental Health Trust where the study was carried out, the SOAS-R has been used since 2007 as a standard tool for nurses to log incidents and medical incident reports, both in inpatient and outpatient settings. 

Question # 30 Do the patients have both MID/BIF and co-morbid mental health conditions? How would the results take into account that the patient may display aggressive behaviours not because of their cognitive functioning and rather their mental health conditions? 

Response # 30 This was a study in a Mental Health Service. Patients were referred to the service to be treated for psychiatric illnesses, with medication or psychotherapy. Our main issue is that more knowledge of MID or BIF may lead to taking the cognitive ability of the patient into account in treating the illness. MID and BIF were determined with the SCIL, but all patients had one psychiatric illness or another, as may be seen in table 2. In the results, we of course take the association of both into account when looking for the association between MID/BIF and aggression. For this reason, in the logistic regression and in the Poisson regression, MID and BIF are included together with the demographic characteristics and the diagnoses. All are included in the regression models. 

Question # 31 In the results table (Table 2 and 3) – how is developmental disorder defined? Some developmental disorders may include symptoms of MID/BIF, how will that affect the results? 

Response # 31 As stated above, it was a study in a Mental Health Service. Patients could have more than one diagnosis. Developmental disorders predominantly concerned the ICD-10 F8 diagnoses and include the next ICD-10 codes: F81.0; F81.2; F81.8; F81.9; F82; F80.1; F80.2; F80.0; F80.9; F84.0; F84.2; F84.3; F84.5; F84.9. Also, the next DSM-IV tr codes were included in this category: 315.00; 315.1; 315.2; 315.9; 315.4; 315.31; 315.32; 315.39; 307.9; 299.00; 299.80; 299.10; 299.80; 299.80. 

Even though one might expect some association between MID/BIF and developmental disorders, this was only partly the case, such as for Autism and Asperger syndrome. On the whole, the association between developmental disorder and MID/ BIF was not remarkable. To investigate this, we calculated a % agreement and a Kappa. In BIF the % agreement was 71%, and in MID it was 70%. The Kappa’s were below 0.10, the correlations as well. We investigated collinearity while performing the regression analyses, which showed no issues. Consequently we feel the concern of the reviewer is addressed in these extra analyses. As it does not concern the focus of the study, we did not present these findings in the publication. 

Question # 32 Some patients are staying in long-stay wards – do social factors play a part that may contribute to the aggressive behaviours instead of cognitive impairment. 

Response # 32 We added a line (373) in recommendations for future research because we do not know yet if or how this plays a role: We also do not know the influence of social factors between patients within ward settings, as the current study did not look into that level of detail. 

Question # 33 How does screening patients for ID support the prevention of aggressive incidents as mentioned in clinical implications? 

Response # 33 We added lines 362-365: The SCIL can be helpful in psychiatric care [29,45] and give a quick first impression. Treatment and support to the needs can then be adapted according to the SCIL category. As such, avoid over-demanding and stress. 

Question # 34 Do the staff’s attitudes towards people with ID play a part to detect and prevent aggressive incidents? 

Response # 34 Yes indeed, this was addressed in line 364: 

The staff’s attitudes towards people with ID may also play a part to detect and prevent aggressive incidents. 

Question # 35 As the staff interact with the patients for a long period of time, it is possible that their relationship with patients may influence their ratings on the scales, would it become a discriminatory tool to segregate people with suspected ID and those without?

Response # 35 We added line 186: According to the questionnaire instructions, the SCIL was administered by a person not involved in the treatment. So, this concern was not the case.

---

## [Decision Letter · Decision Letter 1]

7 Jun 2022

PONE-D-21-40785R1Aggressive behaviour of psychiatric patients with Mild and Borderline Intellectual Disabilities in general Mental Health CarePLOS ONE

Dear Dr. Nieuwenhuis,

I have received the comments from one reviewer on your revised paper and I have read the revision myself. The revision has resulted in a much improved manuscript. I would be happy to accept your paper pending some very minor revisions. The reviewer has provided concrete suggestions where the paper may be revised. I agree with the suggestion to use person-first language. 

We look forward to receiving your revised manuscript.

Kind regards,

Robert Didden

Academic Editor

PLOS ONE

Journal Requirements:

Reviewers' comments:

Reviewer's Responses to Questions

**Comments to the Author**

1. If the authors have adequately addressed your comments raised in a previous round of review and you feel that this manuscript is now acceptable for publication, you may indicate that here to bypass the “Comments to the Author” section, enter your conflict of interest statement in the “Confidential to Editor” section, and submit your "Accept" recommendation.

Reviewer #1: (No Response)

2. Is the manuscript technically sound, and do the data support the conclusions?

Reviewer #1: Yes

3. Has the statistical analysis been performed appropriately and rigorously? 

Reviewer #1: Yes

4. Have the authors made all data underlying the findings in their manuscript fully available?

Reviewer #1: Yes

5. Is the manuscript presented in an intelligible fashion and written in standard English?

Reviewer #1: No

6. Review Comments to the Author

Reviewer #1: Thank you for the opportunity to re-review this manuscript. The authors have addressed most of my concerns. Remaining issues are listed below – I have also commented on some new sections of text.

1. Again I would encourage use of person-first language, e.g. changing “MID/BIF patients” (pg 2, line 67) to “people with MID/BIF”; “SMI patients” (page 5, line 137, line 142, pg 6 line 174) to “people with SMI”; “the homeless” (page 6, line 168) to “people experiencing homelessness”; “MID or BIF cases” (page 8, line 253) to “people with MID or BIF”; “MID patients” (page 8, line 257) to “people with MID”

2. Page 2, line 87, suggest changing “the more severely intellectually impaired patients” to “people with more severe intellectual disability”

3. Page 5, line 135, “Serious Mentally Ill patients’ should read “people with serious mental illness (SMI)”.

4. Again, terms such as mild intellectual disability (pg 2, line 63), borderline intellectual functioning (pg 2, line 63), mental health care (pg 2, line 64; page 4, line 120), challenging behaviour (pg 2, line 73) do not need to have the first letter of each word capitalised.

5. Page 11, line 336: intellectual disability should be abbreviated to ID in the following sentence- “As we can learn from ID studies based on interviews with people with intellectual disability [10, 37], people with ID often experience..”

6. Page 9, line 271: should “Patients with bipolar disorder… were associated with an increased risk of” read “Diagnoses of bipolar disorder… were associated with an increased risk of”?

7. Page 9, line 276-77: check for typographical errors – e.g. “associated to” should read “associated with”

8. Page 9, line 280-285: inconsistent use of “patients who screened positive for BIF or MID were associated with . . . and “MID was associated with”. It would be helpful for the reader to be consistent. I would also recommend this section be reviewed by a statistician or biostatistician to ensure the results are appropriately differentiated in text from those of the Poisson regression. E.g. if logistic regression would be determining risk for having at least one aggressive incident, while the Poisson would look at the number of incidents.

9. Page 11, line 339: if using the term challenging behaviour then this should be used here as well – behaviours that challenge is an alternative expression preferred by some.

10. Page 11, lines 356-360- The authors may wish to include some citations for statements included here.

E.g. for recommendations regarding the prevention and management of behaviours that challenge: The NICE guidelines here https://www.nice.org.uk/guidance/ng11/resources/challenging-behaviour-and-learning-disabilities-prevention-and-interventions-for-people-with-learning-disabilities-whose-behaviour-challenges-1837266392005.

Regarding mental health services frequently being ill-equipped to meet the needs of people with intellectual disability:

Whittle, E.L.; Fisher, K.R.; Reppermund, S.; Lenroot, R.; Trollor, J. Barriers and Enablers to Accessing Mental Health Services for People With Intellectual Disability: A Scoping Review. J. Ment. Health Res. Intellect. Disabil. 2018, 11, 69–102.

Mesa, S.; Tsakanikos, E. Attitudes and self-efficacy towards adults with mild intellectual disability among staff in acute psychiatric wards: An empirical investigation. Adv. Ment. Health Intellect. Disabil. 2014, 8, 79–90.

Evans, E.; Howlett, S.; Kremser, T.; Simpson, J.; Kayess, R.; Trollor, J. Service development for intellectual disability mental health: A human rights approach. J. Intellect. Disabil. Res. 2012, 56, 1098–1109.

Weiss, J.A.; Lunsky, Y.; Gracey, C.; Canrinus, M.; Morris, S. Emergency Psychiatric Services for Individuals with Intellectual Disabilities: Caregivers’ Perspectives. J. Appl. Res. Intellect. Disabil. 2009, 22, 354–362.

Donner, B.; Mutter, R.; Scior, K. Mainstream In-Patient Mental Health Care for People with Intellectual Disabilities: Service User, Carer and Provider Experiences. J. Appl. Res. Intellect. Disabil. 2010, 23, 214–225.

11. Page 11, line 363, the I in ID has been deleted in track changes but should be retained

12. Page 11, like 367- can the authors include a reference to support the link between staff attitudes toward people with intellectual disability and the prevention and detection of aggressive incidents.

13. Page 12, line 370-372, it could be made clearer here that the authors are recommending further work specifically focusing on aggressive behaviour among people with intellectual disability within the mental health service setting, including how this compares to the broader population of people with intellectual disability (if that is what they are recommending).

14. Page 12- line 372- in the sentence starting “for example, we do not yet know much about…” it is not clear whether the authors are referring to mental health diagnosis profiles of their cohort specifically, or mental health service users with intellectual disability more generally. This should be made more clear. Also, MHC should not be abbreviated here as on previous mentions it is expanded as mental health care.

15. Page 12, line 377, suggest including references to support statements about the usefulness of positive behaviour

support.

7. PLOS authors have the option to publish the peer review history of their article (what does this mean?). If published, this will include your full peer review and any attached files.

Reviewer #1: No

---

## [Author Response · Author response to Decision Letter 1]

25 Jun 2022

PONE-D-21-40785R1

Aggressive behaviour of psychiatric patients with Mild and Borderline Intellectual Disabilities in general Mental Health Care

PLOS ONE

1. If the authors have adequately addressed your comments raised in a previous round of review and you feel that this manuscript is now acceptable for publication, you may indicate that here to bypass the "Comments to the Author" section, enter your conflict of interest statement in the "Confidential to Editor" section, and submit your "Accept" recommendation.

Reviewer #1: (No Response)

2. Is the manuscript technically sound, and do the data support the conclusions?

Reviewer #1: Yes

3. Has the statistical analysis been performed appropriately and rigorously? 

Reviewer #1: Yes

4. Have the authors made all data underlying the findings in their manuscript fully available?

Reviewer #1: Yes

5. Is the manuscript presented in an intelligible fashion and written in standard English?

Reviewer #1: No

6. Review Comments to the Author

Reviewer #1: Thank you for the opportunity to re-review this manuscript. The authors have addressed most of my concerns. Remaining issues are listed below – I have also commented on some new sections of text.

Response #1We thank the reviewer for this comment. Also for the new comments. We think the article improved again a lot.

#1. Again I would encourage use of person-first language, e.g. 

a) changing "MID/BIF patients" (pg 2, line 67) to "people with MID/BIF"; "

b) SMI patients" (page 5, line 137, line 142, pg 6 line 174) to "people with SMI"; 

c) "the homeless" (page 6, line 168) to "people experiencing homelessness"; 

d) "MID or BIF cases" (page 8, line 253) to "people with MID or BIF"; 

e) "MID patients" (page 8, line 257) to "people with MID"

#1. response: We have adopted the recommendations. The only difference is that this study was conducted in the mental health care, where we speak of patients. Therefore, we have used patients instead of people throughout the draft. 

2. Page 2, line 87, suggest changing "the more severely intellectually impaired patients" to "people with more severe intellectual disability"

Response 2: we have changed this in ‘patients with more severe intellectual disability’.

3. Page 5, line 135, "Serious Mentally Ill patients' should read "people with serious mental illness (SMI)".

Response 3: Again we changed this, but in patients with serious mental illness

4. Again, terms such as mild intellectual disability (pg 2, line 63), borderline intellectual functioning (pg 2, line 63), mental health care (pg 2, line 64; page 4, line 120), challenging behaviour (pg 2, line 73) do not need to have the first letter of each word capitalised.

Response 4: Thank you for being so precise. We changed this all.

5. Page 11, line 336: intellectual disability should be abbreviated to ID in the following sentence- "As we can learn from ID studies based on interviews with people with intellectual disability [10, 37], people with ID often experience.."

Response 5: Again thank you for being so precise. We changed this.

6. Page 9, line 271: should "Patients with bipolar disorder… were associated with an increased risk of" read "Diagnoses of bipolar disorder… were associated with an increased risk of"?

Response 6: We changed this.

7. Page 9, line 276-77: check for typographical errors – e.g. "associated to" should read "associated with"

Response 7: Most were "associated with". With search and change we detected only one "associated to "in the whole article, and corrected this.

8. Page 9, line 280-285: inconsistent use of "patients who screened positive for BIF or MID were associated with . . . and "MID was associated with". It would be helpful for the reader to be consistent. I would also recommend this section be reviewed by a statistician or biostatistician to ensure the results are appropriately differentiated in text from those of the Poisson regression. E.g. if logistic regression would be determining risk for having at least one aggressive incident, while the Poisson would look at the number of incidents.

Response 8: 

An experienced statistician was one of the authors (Noorthoorn) and carefully reviewed this section. We substantially reformulated the complete section, using for the logistic regression ‘being at risk’ in stead of the more vague ‘associated with’. With respect to the poisson regression we presented the beta (β). 

Logistic regression 

 The logistic regression analysis showed that patients who screened positive for BIF (OR=2.00, p=0.003) or MID (OR 2.89, p<0.001) were more at risk to show aggressive incidents, as well as the patients with the diagnoses bipolar disorder (OR 3.07, p<0.001), schizophrenia (OR 2.75, p<0.001), and a low GAF (OR 1.72, p=0.005). Logistic regression analysis with physical aggression as outcome showed that patients with MID (OR 2.50, p<0.001), a bipolar disorder (OR 3.13, p=0.007) or schizophrenia (OR 4.04, p<0.001) were more at risk to show aggressive incidents. 

Poisson regression 

 These findings were underlined by the poisson regression of the number of physical aggression incidents per patient. This showed anxiety disorder (β=0.62, p<0.001), bipolar disorder (β=1.63, p<0.001), schizophrenia (β=1.12, p<0.001), developmental disorder (β)=0.69, p<0.001), drug abuse disorder (β=1.18, p<0.001) and a SCIL below 15 (β=0.61, p<0.001) were all related to more incidents. 

9. Page 11, line 339: if using the term challenging behaviour, then this should be used here as well – behaviours that challenge is an alternative expression preferred by some.

Response 9: We changed this to challenging behaviour.

10. Page 11, lines 356-360- The authors may wish to include some citations for statements included here.

Response 11:

Thank you for offering some references. We included a reference to The NICE guidelines and to the first two references (Whittle et al and Mesa et al. We did not citate because, in our opinion, this is beyond the scope of this article.

E.g. for recommendations regarding the prevention and management of behaviours that challenge: The NICE guidelines here https://www.nice.org.uk/guidance/ng11/resources/challenging-behaviour-and-learning-disabilities-prevention-and-interventions-for-people-with-learning-disabilities-whose-behaviour-challenges-1837266392005.

Regarding mental health services frequently being ill-equipped to meet the needs of people with intellectual disability:

Whittle, E.L.; Fisher, K.R.; Reppermund, S.; Lenroot, R.; Trollor, J. Barriers and Enablers to Accessing Mental Health Services for People With Intellectual Disability: A Scoping Review. J. Ment. Health Res. Intellect. Disabil. 2018, 11, 69–102.

Mesa, S.; Tsakanikos, E. Attitudes and self-efficacy towards adults with mild intellectual disability among staff in acute psychiatric wards: An empirical investigation. Adv. Ment. Health Intellect. Disabil. 2014, 8, 79–90.

We did not include the next publications suggested: 

Evans, E.; Howlett, S.; Kremser, T.; Simpson, J.; Kayess, R.; Trollor, J. Service development for intellectual disability mental health: A human rights approach. J. Intellect. Disabil. Res. 2012, 56, 1098–1109.

Weiss, J.A.; Lunsky, Y.; Gracey, C.; Canrinus, M.; Morris, S. Emergency Psychiatric Services for Individuals with Intellectual Disabilities: Caregivers' Perspectives. J. Appl. Res. Intellect. Disabil. 2009, 22, 354–362.

Donner, B.; Mutter, R.; Scior, K. Mainstream In-Patient Mental Health Care for People with Intellectual Disabilities: Service User, Carer and Provider Experiences. J. Appl. Res. Intellect. Disabil. 2010, 23, 214–225.

11. Page 11, line 363, the I in ID has been deleted in track changes but should be retained

Response 11: Thank you for your attention. We changed this.

12. Page 11, like 367- can the authors include a reference to support the link between staff attitudes toward people with intellectual disability and the prevention and detection of aggressive incidents.

Response 12: We included the next references to support this link: 

van den Bogaard KJHM, Nijman HLI, Embregts PJCM. Attributional styles of support staff working with people with intellectual disabilities exhibiting challenging behaviour. J Appl Res Intellect Disabil. 2020 May;33(3):465-474.

Knotter MH, Wissink IB, Moonen XM, Stams GJ, Jansen GJ. Staff's attitudes and reactions towards aggressive behaviour of clients with intellectual disabilities: a multi-level study. Res Dev Disabil. 2013 May;34(5):1397-407.

13. Page 12, line 370-372, it could be made clearer here that the authors are recommending further work specifically focusing on aggressive behaviour among people with intellectual disability within the mental health service setting, including how this compares to the broader population of people with intellectual disability (if that is what they are recommending).

Response 13:

Thank you for your comment. The lines 370-372 concern not further work (etc.) but concerns about giving staff and nurses a better opportunity for personal tailormade treatment and attitude toward patients with ID. We explained in lines 376-381 more precisely what kind of future research is recommended.

We changed the sentence and added an extra sentence ( and reference) to be more clear;

 " In mental health care, future studies should examine the causes and reasons for aggression incidents in patients with MID/BIF and learn more about how these patients differ or resemble those of people in ID studies. For example, we do not yet know much about the mental problems in general in the patient group with ID in mental health care and how this affects behaviour, including how this compares to the broader population of people with intellectual disability and mental health problems [..]. 

14. Page 12- line 372- in the sentence starting "for example, we do not yet know much about…" it is not clear whether the authors are referring to mental health diagnosis profiles of their cohort specifically or mental health service users with intellectual disability more generally. This should be made more clear. Also, MHC should not be abbreviated here as in previous mentions it is expanded as mental health care.

Response 14: 

As explained in our response 13, we tried to be more clear.

MCH has been changed in mental health care.

15. Page 12, line 377, suggest including references to supporting statements about the usefulness of positive behaviour

support.

Response 15: we added the next reference: Lewis N, Reynolds E, Vale M, Keenan E, Hartland A, Haines M, Davies BE. An evaluation of positive behavioural support implemented within an intensive community support service for people with learning disabilities who present with behaviours that challenge. J Intellect Disabil. 2021 Sep;25(3):348-356.

7. PLOS authors have the option to publish the peer review history of their article (what does this mean?). If published, this will include your full peer review and any attached files.

Do you want your identity to be public for this peer review? For information about this choice, including consent withdrawal, please see our Privacy Policy.

Reviewer #1: No

---

## [Decision Letter · Decision Letter 2]

4 Jul 2022

PONE-D-21-40785R2Aggressive behaviour of psychiatric patients with Mild and Borderline Intellectual Disabilities in general Mental Health CarePLOS ONE

Dear Dr. Nieuwenhuis,

I have received the comments from the reviewer on your revised paper. I would be happy to accept your paper pending some very minor revisions. Please take a close look at the reviewer's comments and suggestions which are very minor and textual. 

We look forward to receiving your revised manuscript.

Kind regards,

Robert Didden

Academic Editor

PLOS ONE

Journal Requirements:

Reviewers' comments:

Reviewer's Responses to Questions

**Comments to the Author**

1. If the authors have adequately addressed your comments raised in a previous round of review and you feel that this manuscript is now acceptable for publication, you may indicate that here to bypass the “Comments to the Author” section, enter your conflict of interest statement in the “Confidential to Editor” section, and submit your "Accept" recommendation.

Reviewer #1: (No Response)

2. Is the manuscript technically sound, and do the data support the conclusions?

Reviewer #1: Yes

3. Has the statistical analysis been performed appropriately and rigorously? 

Reviewer #1: Yes

4. Have the authors made all data underlying the findings in their manuscript fully available?

Reviewer #1: Yes

5. Is the manuscript presented in an intelligible fashion and written in standard English?

Reviewer #1: No

6. Review Comments to the Author

Reviewer #1: Thank you for sending this manuscript for re-review. The edits related to the statistical analyses were very helpful. I would just suggest the statistician on the author team also review the abstract to ensure the results are represented appropriately there. I noted that the track changes in the document denote changes made in the first round of revision while the most recent edits are not tracked. I did not review the manuscript thoroughly as I think there may be some issues with the file:

1. Typographical errors that the authors said have been fixed have not been; e.g. serious mentally ill on pg 15 should read patients with serious mental illness; the I in ID is still deleted on page 11, line 363; MHC is still abbreviated on page 12, line 373. I did not check the rest and recommend the authors ensure their response matches the manuscript document.

2. Response 13 (re lines 30-381) does not appear to be incorporated into the manuscript. Nonetheless I still suggest the authors reword this as it is still not clear to me what they are recommending, and suggest the authors ensure use of people first language (i.e. studies with people with ID, not people in ID studies), and also to be consistent with abbreviating intellectual disability to ID.

Also, some minor typographical issues in the abstract:

3. Some words are incorrectly capitalized in the abstract; mild intellectual disability, borderline intellectual functioning, mental health care, behaviour, odds (or if this is to be capitalized then ratio should be capitalized as well), bipolar disorder. Also, “was higher in assumed MID” should read “in patients with assumed MID”

7. PLOS authors have the option to publish the peer review history of their article (what does this mean?). If published, this will include your full peer review and any attached files.

Reviewer #1: No

---

## [Author Response · Author response to Decision Letter 2]

20 Jul 2022

Reviewer #1: Thank you for sending this manuscript for re-review. The edits related to the statistical analyses were very helpful. I would just suggest the statistician on the author team also review the abstract to ensure the results are represented appropriately there. I noted that the track changes in the document denote changes made in the first round of revision while the most recent edits are not tracked. I did not review the manuscript thoroughly as I think there may be some issues with the file:

#1a: We added three brief sections to the abstract highlighting the main findings of the analyses which were added in the process of the rebuttal. On page 2 line 46 we added: and poisson. On page 2 line 52 we added the confidence interval of the OR for BIF: OR 4.20, 95% CI 2.45-7.22. And on page 2 line 55-56 we added: Poisson regression underlined these findings, showing a SCIL below 15 (β=0.61, p<0.001) was related to more incidents. We chose not to discuss all findings of the poisson regression in the same detail as in the results section, as the poisson regression was primarily used as a confirmation of the findings of the logistic regression. 

1. Typographical errors that the authors said have been fixed have not been; e.g. serious mentally ill on pg 15 should read patients with serious mental illness; the I in ID is still deleted on page 11, line 363; MHC is still abbreviated on page 12, line 373. I did not check the rest and recommend the authors ensure their response matches the manuscript document.

#1b. Thank you very much for your comments on typographical errors. Indeed there have been some issues with the file but we also find again some errors scattered throughout the text. These have been corrected. 

2. Response 13 (re lines 30-381) does not appear to be incorporated into the manuscript. Nonetheless I still suggest the authors reword this as it is still not clear to me what they are recommending, and suggest the authors ensure use of people first language (i.e. studies with people with ID, not people in ID studies), and also to be consistent with abbreviating intellectual disability to ID.

#2 Thank you for persisting in using first language. We have again changed a number of sentences

Also, some minor typographical issues in the abstract:

3. Some words are incorrectly capitalized in the abstract; mild intellectual disability, borderline intellectual functioning, mental health care, behaviour, odds (or if this is to be capitalized then ratio should be capitalized as well), bipolar disorder. Also, “was higher in assumed MID” should read “in patients with assumed MID”

#3 Thank you for your comment. We have aligned the use of capital letters throughout the article as much as possible

---

## [Editor Report · Decision Letter 3]

21 Jul 2022

Aggressive behaviour of psychiatric patients with Mild and Borderline Intellectual Disabilities in general Mental Health Care

PONE-D-21-40785R3

Dear Dr. Nieuwenhuis,

We’re pleased to inform you that your manuscript has been judged scientifically suitable for publication and will be formally accepted for publication once it meets all outstanding technical requirements.

Kind regards,

Robert Didden

Academic Editor

PLOS ONE
---

## [Editor Report · Acceptance letter]

19 Sep 2022

PONE-D-21-40785R3 

Aggressive behaviour of psychiatric patients with Mild and Borderline Intellectual Disabilities in general Mental Health Care 

Dear Dr. Nieuwenhuis:

I'm pleased to inform you that your manuscript has been deemed suitable for publication in PLOS ONE. Congratulations! Your manuscript is now with our production department. 

Kind regards, 

on behalf of

Professor Robert Didden 

Academic Editor

PLOS ONE